# Multi-View Analysis of High-Resolution Geomorphic Features in Complex Mountains Based on UAV–LiDAR and SfM–MVS: A Case Study of the Northern Pit Rim Structure of the Mountains of Lufeng, China

**Rui Bi [1], Shu Gan [1,2,*], Xiping Yuan [2,3], Raobo Li [1], Sha Gao [1], Min Yang [3], Weidong Luo [1] and Lin Hu [1]**

[1] School of Land and Resources Engineering, Kunming University of Science and Technology, Kunming 650093, China

[2] Plication Engineering Research Center of Spatial Information Surveying and Mapping Technology in Plateau and Mountainous Areas Set by Universities in Yunnan Province, Kunming 650093, China

[3] Key Laboratory of Mountain Real Scene Point-Cloud Data Processing and Application for Universities in Yunnan Province, West Yunnan University of Applied Sciences, Dali 671006, China

*  Correspondence: gs@kust.edu.cn; Tel.: +86-135-7703-2539

**Abstract:** Unmanned aerial vehicles (UAVs) and light detection and ranging (LiDAR) can be used to analyze the geomorphic features in complex plateau mountains. Accordingly, a UAV–LiDAR system was adopted in this study to acquire images and lidar point-cloud dataset in the annular structure of Lufeng, Yunnan. A three-dimensional (3D) model was constructed based on structure from motion and multi-view stereo (SfM–MVS) in combination with a high-resolution digital elevation model (DEM). Geomorphic identification, measurement, and analysis were conducted using integrated visual interpretation, DEM visualization, and geographic information system (GIS) topographic feature extraction. The results indicated that the 3D geomorphological visualization and mapping were based on DEM, which was employed to identify the dividing lines and ridges that were delineated of the pit rim structure. The high-resolution DEM retained more geomorphic detail information, and the topography and the variation between ridges were analyzed in depth. The catchment and ponding areas were analyzed using accurate morphological parameters through a multi-angle 3D visualization. The slope, aspect, and topographic wetness index (TWI) parameters were analyzed through mathematical statistics to qualitatively and accurately analyze the differences between different ridges. This study highlighted the significance of the UAV–LiDAR high-resolution topographic measurements and the SfM–MVS 3D scene modelling in accurately identifying geomorphological features and conducting refined analysis. An effective framework was established to acquire high-precision topographic datasets and to analyze geomorphological features in complex mountain areas, which was beneficial in deepening the research on numerical simulation analysis of geomorphological features and reveal the process evolution mechanism.

**Keywords:** UAV; LiDAR; pit-rim structure; geomorphic feature; SfM–MVS; visualization

## 1. Introduction

Geomorphology refers to the formation of a wide variety of undulating forms on the earth's surface since internal and external geological forces are continuously being shaped. It is a vital factor in understanding and analyzing the distribution of geological features of the natural geographic environment and geological structures [1,2]. The analysis of geomorphic features takes on a critical significance to the research of natural environment change [3], land use type distribution [4], disaster identification [5], and geological tectonic movement [6].

The conventional method of geomorphological surveying and feature analysis is to use GIS to conduct a considerable amount of manual field investigation and statistical

analysis [7,8]. Furthermore, the total station is adopted to observe small areas and lower elevation geomorphological patterns [9]. Alternatively, global navigation satellite system (GNSS) technology has been employed to obtain topographic data, before geomorphic features are classified and mapped with the use of the obtained data [10]. Although the above methods exhibit high measurement accuracy and have been extensively used, they are time-consuming and laborious. Notably, for complex geomorphic features where survey operations are risky and inefficient, detailed information regarding the features may be missing [11], and the survey results may not indicate the current situation.

As space–sky–ground remote sensing technology has been updated, and a wide variety of data acquisition methods have been proposed, high spatial resolution and multi-source remote sensing data provide considerable sources for the carrying out of high-quality and efficient geomorphological surveys and the acquiring of detailed geomorphological information [12]. With the fusion of multi-source spectral data, large-scale and long-time series surveys and quantitative evolutionary analysis can be achieved [13]. Synthetic aperture radar (SAR) images are adopted to monitor geographical features (e.g., landslides [14], saline lands [15], and rock glaciers [16]). However, optical remote sensing images and radar images are limited by natural environmental factors (e.g., imaging time, spatial resolution, cloudiness, different orbits, and bands), significantly reducing the regularity and quality of data acquisition [17,18]. The future development trend is toward more refined and accurate geomorphological feature analysis, so that the question as to how to acquire high-quality geomorphic data takes on critical significance.

Over the past few years, UAVs have constructed 3D models using their acquired high-resolution images with SfM–MVS and have identified and extracted geographical features using the derived DEMs [19,20]. Li, et al. [21] obtained paleoseismic geomorphological images using a UAV and quantitatively analyzed paleoseismic offsets using DEM. Granados-Bolanos, et al. [22] obtained high-resolution images of the volcanic and classified landforms by combining the digital orthophoto map (DOM) and the DEM. Gong, et al. [23] studied the correlation between the development process of an erosion gully and microscopic topographic factors using obtained multi-period centimeter-level erosion gully DOMs and DEMs. DEM has been widely used as a vital source for a wide range of geomorphological feature analyses. However, in a complex geomorphic environment with high vegetation coverage or large topographic relief, the images obtained by conventional UAV route planning methods cannot indicate the real surface, and the interpolated DEM exhibit sharp points [24], thus decreasing the accuracy of geomorphic feature analysis.

LiDAR remote sensing technology uses active laser transmitters and receivers to accurately acquire elevation data [25]. It is noteworthy that terrestrial LiDAR scanning (TLS) has become one of the most accurate and reliable methods of constructing high-resolution DEMs on complex landforms [26,27]. Although it can acquire high-precision data, the instrument should be repeatedly moved for scanning, as some areas are not easily accessible by the laser. The periodic analysis of the amount of geomorphic change may lead to reduced model quality and may introduce various errors, thus resulting in significant differences in analysis results and associated challenges [28].

The UAV-based airborne LiDAR (ALS) technology can provide more detailed 3D surface models [29]. Jagodnik et al. [30] used ALS to identify lithological and geomorphological processes in large-scale valley areas for engineering geological mapping. He et al. [31] analyzed the secondary debris flow disaster chain caused by landslides using the high-resolution DEM from the airborne LiDAR. Sare et al. [32] acquired fault cliffs and fault-related geomorphological data, extracted semi-automatic fault zone features, and conducted feature analysis. High-resolution DEMs have been extensively used in various geomorphological surveys and analyses. LiDAR can more flexibly adapt to complex topographic environments and compensate for the inability of optical remote sensing images to truly indicate the ground surface.

However, the monolithic and insufficient quality data cannot provide the basic analysis required for high-precision large-area geomorphological feature detection and analysis in

highland and mountainous areas with significant vertical drop and a complex topographic environment. The fusion of the UAV–LiDAR technology not only solves the problem of insufficient data quality due to the natural environment but also makes up for the identification and analysis of incomplete geographical features with multi-source data results. UAVs can provide a wide range of comprehensive views from images, generate highly detailed topographic information, rebuild the true shape of the geomorphology [33], optimize conventional survey methods, and increase the efficiency and accuracy of survey results. Additionally, 3D point-cloud data obtained by LiDAR technology can remove the effect of vegetation and can quickly and accurately construct DEMs since climate and vegetation significantly affect the flight position of UAVs, which further affects the accuracy of UAV images and DEMs [34]. High-precision DEMs are capable of collecting very useful information to describe the geomorphology. Moreover, they are critical to facilitating subtle geographical changes and the accurate identification of different geographical types. DEMs take on a certain scientific significance and practical effect in how to more effectively detect the special annular structure of geomorphological data in the mountainous environment of central Yunnan plateau and in the identification and quantitative analysis of geomorphological features from multiple perspectives.

In this study, the annular structure geomorphology of the Dinosaur Valley of Lufeng City, Chuxiong, Yunnan was taken as the research area, and the complex pit rim structure was selected as the research object. High-resolution images and high-quality point-cloud data were obtained automatically using the UAV–LiDAR system. The 3D model was constructed using the SfM–MVS technology, and the LiDAR point-cloud was processed to build the high-resolution DEM. High-precision topographic survey data and numerical simulation of 3D topographic scenes were comprehensively utilized. Multi-angle, qualitative, and quantitative geographical feature was identified, measured, and analyzed through visual interpretation, DEM visualization, GIS topographical feature extraction, and analysis. Furthermore, efficient and high-quality technology and methods were provided in this study.

## 2. Materials and Methods

### 2.1. Study Area

The research area was located in the annular structure of the Dinosaur Valley on the southwest side of Lufeng City, Chuxiong, Yunnan (Figure 1a). The annular structure (Figure 1b) was nearly 3 km wide from east to west, about 10 km long from north to south, and nearly 4 km in diameter. The inside was a huge depression, surrounded by steep mountains, and it formed a radial striped ridge, in sharp contrast with the central topography. The complex pit rim at the highest entrance in the northeast was selected for the experiment (Figure 1c). It can be a typical area for exploration, measurement, and analysis.

### 2.2. Data Collection

#### 2.2.1. Airborne LiDAR Point-Cloud and UAV Images

The survey was carried out under the condition of light breeze in the afternoon to prevent changes in the natural environment affecting the flight status and data quality of UAV. The LiAir VH2 UAV–LiDAR system, which has integrated light and small LiDAR systems, a global navigation satellite system, and a digital camera, was installed on a DJI M300 (Figure 2a). This system was equipped with the Livox Avia LiDAR and Sony A5100 camera. This LiDAR exhibited a wavelength of 905 nm, a maximum scanning frequency of 240 kHz/s, and a data acquisition rate of 720,000 points/s under the three-echo observation at a scanning frequency of 160 kHz/s. The field angles were 70.4° and 4.5°. The Sony A5100 camera was equipped with a 16 mm fixed-focus lens with an effective pixel count of 24.3 million. The above parameters are listed in Table 1.

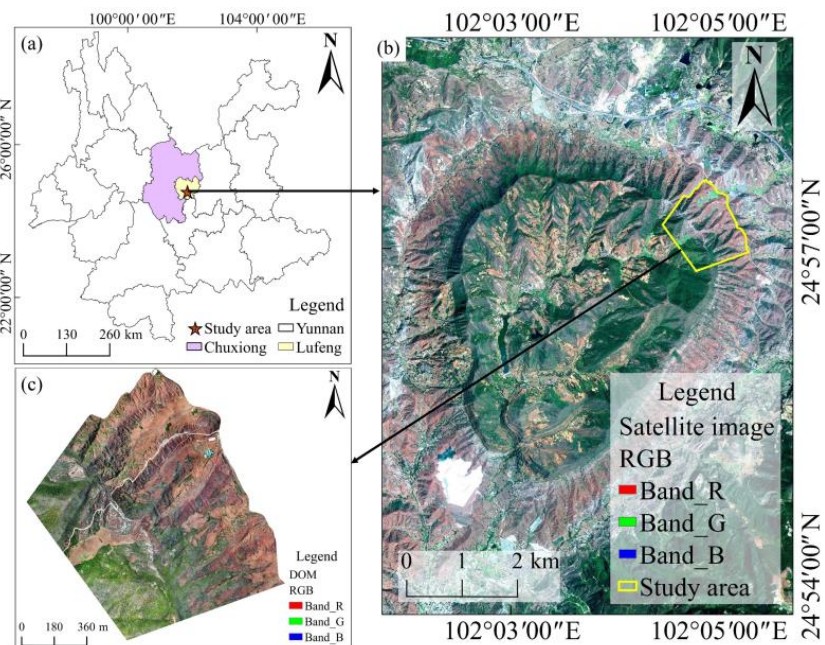

**Figure 1.** (**a**) The location of the study area, (**b**) the satellite image of the annular structure, and (**c**) the complex pit rim geomorphology.

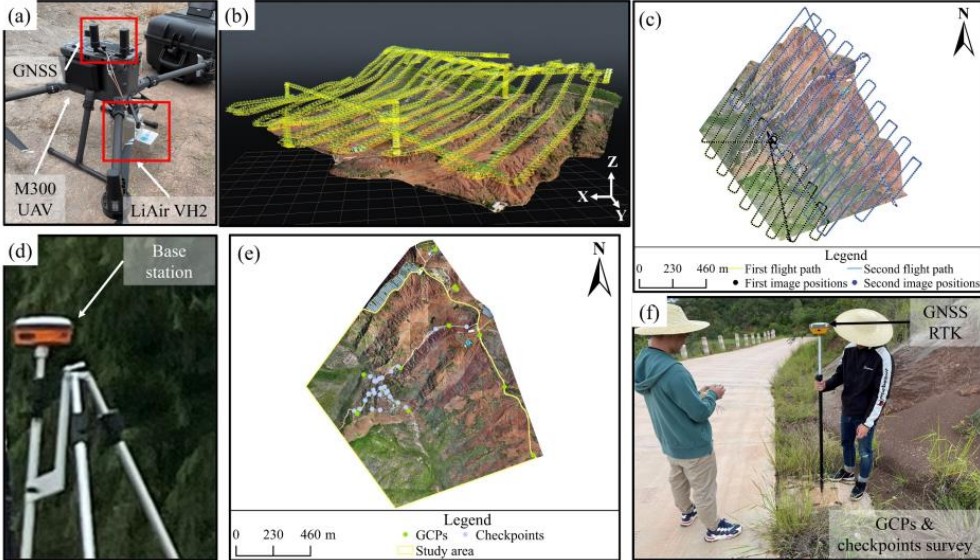

**Figure 2.** (**a**) The LiAir VH2 LiDAR system, (**b**) the UAV flight trajectory, (**c**) the ground-based flight route, (**d**) GNSS base station, (**e**) GCPs and checkpoints layout, and (**f**) GNSS measurements.

In this study, a ground-based flight route planning method was adopted (Figure 2b) to avoid problems (e.g., low flight efficiency, lack of images, data redundancy, image distortion, and low overlap) caused by changes in topography relief [35]. A DEM with a resolution of 12.5 m served as the basic data. The flight altitude was set at 100 m relative to the surface, the flight speed was 8 m/s, the airborne LiDAR side overlap was 35%, the forward overlap was 90%, and an area of 1.2 km$^2$ was scanned twice through ortho-view and three-echo observation (Figure 2c). In addition, the scanning point density was 213.82 points/m$^2$, and 2240 images were obtained. Moreover, the ground base station data with the coordinate system CGCS 2000 was acquired in real time using the Qianxun SR3 Pro RTK equipment (Figure 2d), which provided an accurate coordinate reference for LiDAR data interpretation. The device exhibited static horizontal accuracy of $\pm 8$ mm + 1 ppm and elevation accuracy of $\pm 5$ mm + 1 ppm. Table 2 lists the UAV flight parameters and collected data statistics.

**Table 1.** The UAV flight platform, LiDAR system, and digital camera parameters.

| UAV Specifications | | Airborne LiDAR Specifications | | Camera Specifications | |
|---|---|---|---|---|---|
| Aircraft | M300 RTK | LiDAR system | Livox Avia | Sensor | Sony A5100 |
| Weight | 6300 g | Laser wavelength | 905 nm | | |
| Max flight speed | 50 km/h (Positioning) 58 km/h (Attitude) | Laser pulse repetition rate | 240 kHz | Sensor format | 23.5 × 15.6 mm |
| Max flight time | 55 min | Maximum echo number | 3 | | |
| Max take-off altitude | 5000 m | Maximum scan speed | 720,000 points/second | Focal length | 16 mm |
| Satellite positioning systems | GPS + GLONASS + BeiDou + Galileo | Satellite positioning systems | GLONASS + BeiDou + Galileo | | |
| GNSS positioning accuracy (RTK Fixed) | 1 cm + 1 ppm (Horizontal) 1.5 cm + 1 ppm (Vertical) | Field of view | 70.4° (Horizontal) 4.5° (Vertical) | Image resolution | 6000 × 4000 |
| | | Scan pattern | Repeat scan | | |

**Table 2.** The UAV flight parameters and collected data information statistics.

| UAV Flight Parameters | | LiDAR Data | | Image Data | |
|---|---|---|---|---|---|
| Flight altitude | 100 m | Number of scanning routes | 36 | Number of images | 2240 |
| Ground Sampling Distance (GSD) | 2.34 cm/px | Mean density of points | 213.82 points/m$^2$ | Image forward overlap | 90% |
| Airborne LiDAR side overlap | 35% | | | | |
| Camera angle | −90° | Number of points | 3,258,544,456 | Image side overlap | 80% |
| Flight time | 1 h 48 min | | | | |

### 2.2.2. GCPs and Checkpoints

The significant feature points inside and outside the pit rim structure were selected as the ground control points (GCPs) and the checkpoints, respectively. The GCPs were adopted to construct the 3D model, and the LiDAR point-cloud data results and the measurement accuracy of the 3D model were comprehensively analyzed using the checkpoints. Fourteen GCPs and 26 checkpoints were acquired using the Hi-Target V90 RTK connected to the continuously operating reference stations (CORS) network (Figure 2e,f) with a CGCS 2000. It achieved a horizontal accuracy of ±8 mm + 1 ppm and an elevation accuracy of ±15 mm + 1 ppm.

### 2.3. Data Processing

#### 2.3.1. LiDAR Point-Cloud Processing

The raw LiDAR data were pre-processed using LiGeoreference software. First, the original trajectory was segmented. Subsequently, the acquired images and GNSS base station data were added to process and solve. Lastly, the calculated point-cloud data were colored, and 36 bands were obtained.

The LiDAR point-cloud obtained by the solution was further processed in accordance with the same processing flow (e.g., removing repeated data between flight bands, denoising, filtering, and extracting ground points). The raw data were denoised using a statistical outlier detection method that identified outliers by determining whether the distance between a point and its *k* neighboring points exceeded a threshold of *avg. + std.* (*avg.* and *std.* represent the average and standard deviation of distances between points and their *k* neighboring points, respectively) [36]. The ground points were extracted using the improved progressive TIN densification (IPTD) algorithm proposed by Zhao, et al. [37]. Figure 3 presents the LiDAR data processing flow in the black box.

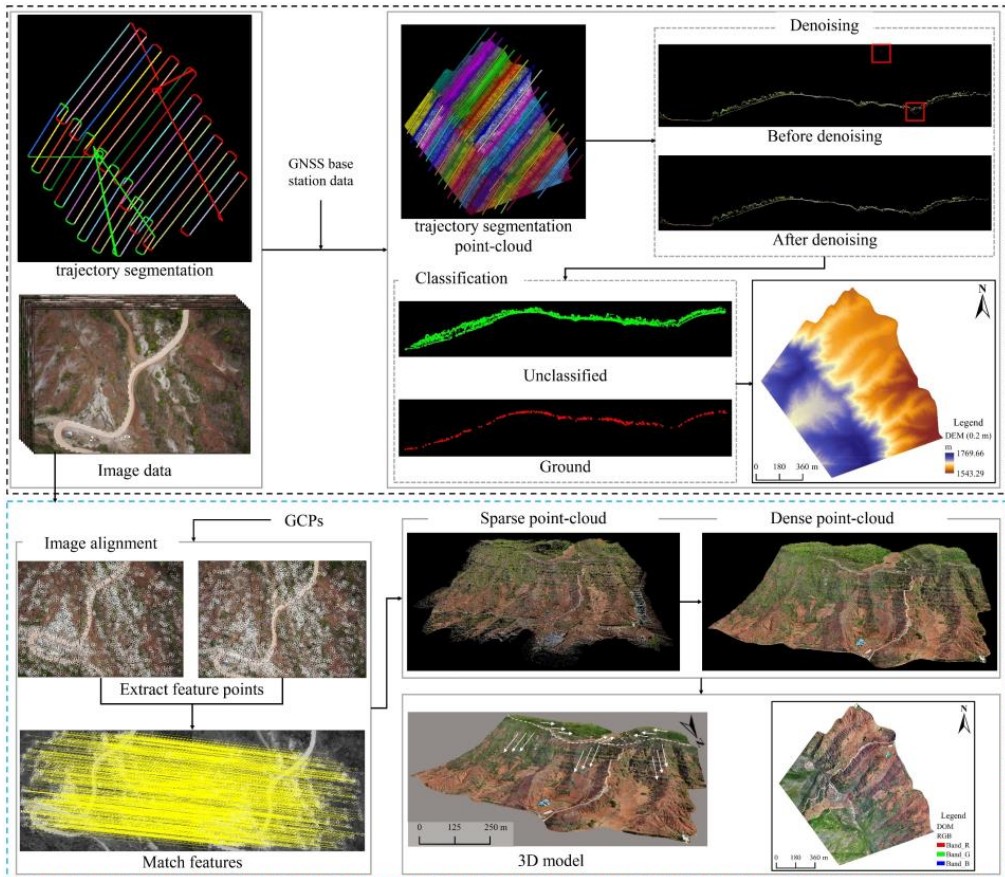

**Figure 3.** LiDAR point-cloud processing and 3D model construction (the black box represents the LiDAR point-cloud data processing process, and the blue box represents the 3D model construction process).

In ArcGIS 10.7, the ground point-cloud was interpolated using the inverse distance weight interpolation (IDW) [38,39] method to obtain a DEM with a resolution of 0.2 m. Furthermore, the DEM's elevation accuracy was checked. The mean elevation deviation reached 3.5 cm, the standard deviation was 4.5 cm, and the root mean square error was 4.4 cm.

### 2.3.2. Construction of 3D Model Based on SfM–MVS

The 3D model was constructed using the ContextCapture Center software integrated with SfM–MVS technology. The SfM technology is capable of solving the spatial and geometric relationship of the target via the movement of the camera from multiple images for model construction [40].

To begin with, in the SfM algorithm, the scale invariant feature transform (SIFT) algorithm [41] was employed to extract the feature points and obtain feature description parameters corresponding to the respective image. Next, image features were matched based on critical points on the overlapped images in combination with image pose and spatial location information, and model parameters (e.g., camera focal length, radial aberration, and tangential aberration). Lastly, a bundle block adjustment [42] was performed to optimize the key point features and camera parameter positions by minimizing the reprojection error between the locations of vital points on the image and the predicted locations [43]. A sparse point-cloud with coordinate and color information was obtained.

The MVS technology iteratively diffused and filtered the sparse point-cloud to obtain a dense point-cloud [44]. Subsequently, the dense point-cloud was segmented into blocks, and the block point-cloud was constructed into an irregular 3D grid. Next, the white film data were acquired from the grid data, and the optimal texture was automatically found

by a triangular grid to achieve automatic optimal texture mapping. Lastly, a 3D model consistent with the coordinate system was obtained. The construction of sparse point-cloud and dense point-cloud data was dependent on the position and attitude information of the image, thus requiring a certain number of GCPs to participate in the calculation, and the model had accurate absolute coordinates [45]. Furthermore, a DOM with 0.05 m resolution was obtained. The 3D model construction process flow is presented in the blue box in Figure 3.

Specifically, the absolute accuracy of the 3D model was analyzed. The root mean square error of the horizontal accuracy and elevation accuracy served as the accuracy assessment index. The accuracy results showed that the horizontal accuracy was 4.4 cm, and the elevation accuracy was 7.9 cm.

### 2.4. Structural Geomorphic Features Analysis Methods

To achieve multi-view and multi-method identification and measurement of geomorphic features, a more comprehensive and clear visualization was performed using DEM by initially recognizing geomorphic features with the use of the DOM information (e.g., color, morphology, and texture). However, in the visual topography analysis of DEM, the change of solar azimuth and altitude [46] will significantly change the visualization of pit rim structural features with significant vertical differences, thus causing omissions and misidentifications. Accordingly, a raster visualization toolbox (RVT) [47] was adopted to adequately generate DEM visualization and analysis results by adjusting the parameters of the visualization technology [48,49] to more comprehensively represent topography morphology and geomorphic features. The visualized topography results were further explained and then analyzed using ArcGIS 10.7.

Based on the visual interpretation, the morphological parameters were examined using ArcGIS and 3D model measurement tools. The difference between the inside and outside topography of the pit rim structure was significant, the dividing line fully indicated the more subtle real surface undulation variation. Thus, the elevation values of the inside and outside longitudinal and transverse sections were extracted using the DEM. The outside ridges and valleys were interlaced, and the valley lines and the ridge lines constitute the boundary of the topographic fluctuation change. By extracting the valley lines and the ridge lines, the watershed and the confluence area could be distinguished to achieve intuitive topography analysis.

The topographic feature parameters (e.g., slope, slope aspect, terrain relief, and TWI) were selected for analysis. The results were determined using the mathematical statistics method. In particular, the type of distribution of the topographic indices can be described using the statistical skewness coefficients. The skewness values > 0 indicate that the distribution frequency tends to show a positive skewness, which means that the regions exhibit increasingly lower values of topography. In contrast, the values < 0 indicate a negatively skewed distribution, which means the regions exhibit increasingly higher values of topographic indexes [50]. The relatively larger skewness values suggest that the frequency distribution of topographic indexes is correlated with an increased proportion of lower topographic index values and vice versa [51]. Slope is capable of affecting the stability of the topography [52], while aspect can affect changes in natural environmental factors (e.g., sunlight, wind, and rainfall) [53]. Topographic relief indicates the difference between the elevation of the highest point and the elevation of the lowest point, thus suggesting the degree of change in the morphology of landform erosion [54]. Slope, aspect, and topographic relief were extracted and then analyzed using ArcGIS 10.7. TWI represents the logarithm of the upper and middle reaches of the topography divided by the slope [55], which is capable of assessing the static soil water content [56]. TWI was implemented in SAGA GIS 8.3.0 of the system for automated geoscientific analyses (SAGA) [57]. All topographic feature parameters were implemented based on the high-resolution DEM.

## 3. Analysis of the Main Geomorphological Features of the Pit Rim Structure

### 3.1. Morphological Parameter Extraction Analysis

A single DOM cannot effectively distinguish and delineate the inside and outside due to the significant variation in the topographic relief. Accordingly, a sky illumination model (SIM) [58] was used to visualize the DEM. A cloudy sky model was built to improve the contrast of the natural brightness of the shadow part (Figure 4a). The SIM results indicate that the natural brightness varied significantly. The dividing line *MM′* between the inside and outside was extracted (Figure 4a). The inside and outside divisions presented a low topography in the middle and high topography at the ends (Figure 4b,c). The trend of elevation change on the right side of the dividing line was higher than the left side, and the mountains on the left side of the dividing line became more rounded. The topography tended to slow down in the middle, and there was significant soil erosion, which was largely distributed in loose red soil.

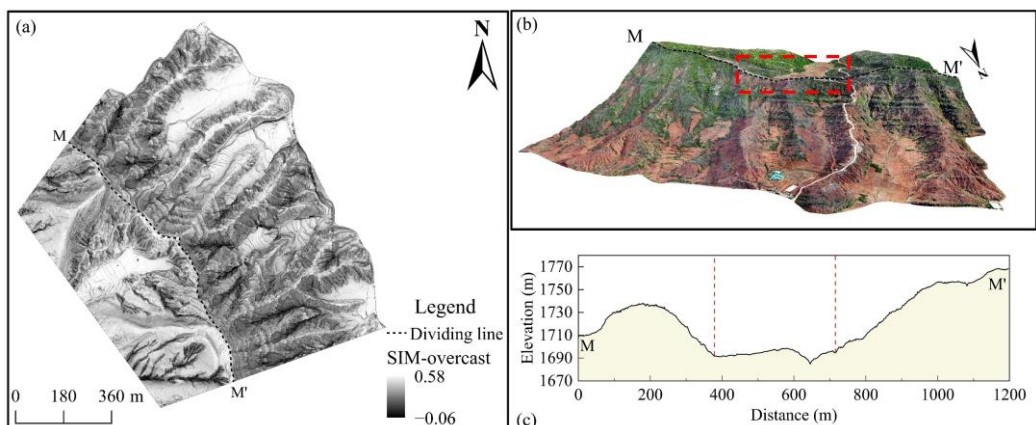

**Figure 4.** (**a**) The dividing line extraction based on SIM-overcast, (**b**) the dividing line display combined with 3D model, and (**c**) the dividing line elevation.

The visualization results of the anisotropic sky-view factor (SVF-A) model [59] were able to provide basic mountain features, especially for ridges that rise on the outside. The ridge part would be more rightly illuminated, for the depression part was less illuminated and had a good outline between steep and flat surfaces for steep topography [60]. Thus, the SVF-A visualization result (Figure 5a), the 3D model, and the actual survey results were combined to divide the outside into ten independent ridges (Figure 5a). Based on the 3D identification results (Figure 5b,c), it could be seen that each ridge extended in a bar from the highest part of the dividing line to the lower part, and some of the ridges diverged in the lower part to form new multiple ridges at their ends (#2, #6 and #8). Specifically, the continuously undulating and changing topography within ridge #7 was considered a ridge.

Morphological parameters can intuitively describe the morphological characteristics of different geographical features [61]. The morphological parameters (e.g., perimeter, area, maximum length, maximum width, L/W ratio, maximum elevation difference, maximum slope, average slope, maximum relief, and average relief) were determined using ArcGIS statistical tools and 3D model measurement tools.

The mean morphological parameters of the ten ridges (Table 3) comprised the perimeter of 1366.92 m, the area of 0.07 km$^2$, the maximum length of 197.39 m, the maximum width of 197.39 m, the L/W ratio of 2.89, the maximum elevation difference of 161.12 m, the maximum slope of 85.19°, the average slope of 32.73°, the maximum relief of 2.85 m, and the average relief of 0.18 m. The #1 and #3 ridges achieved the smallest perimeter and area, they were close to the upper part of the pit rim structure. Ridge #4 exhibited the largest length, smallest mean slope, and lowest mean relief, with a road that connected the inside and outside. Ridge #7 exhibited the maximum width and ridge #9 achieved the maximum L/W ratio with an overall shape that was narrower and more elongated.

Ridge #10 exhibited the largest elevation difference and largest mean slope, and the overall topography was steeper.

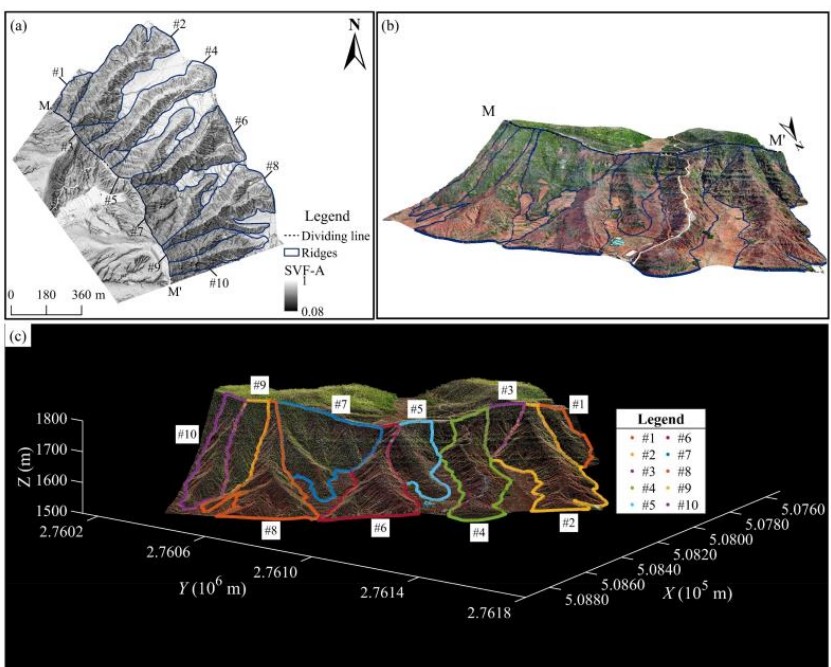

**Figure 5.** (**a**) Extraction of the ridges based on the SVF-A model, (**b**) display of the ridges combined with the 3D model, and (**c**) the 3D recognition result of the ridges.

**Table 3.** Measurement results of morphological parameters of ridges.

| Ridge ID | Perimeter (m) | Area (km²) | Max Length (m) | Max Width (m) | L/W Ratio | Max Elevation Difference (m) | Max Slop (°) | Mean Slop (°) | Max Relief (m) | Mean Relief (m) |
|---|---|---|---|---|---|---|---|---|---|---|
| #1 | 696.50 | 0.02 | 280.72 | 103.24 | 2.72 | 128.76 | 85.53 | 35.01 | 2.75 | 0.20 |
| #2 | 1938.42 | 0.12 | 697.77 | 228.14 | 3.06 | 154.32 | 85.22 | 30.69 | 3.23 | 0.17 |
| #3 | 574.27 | 0.01 | 217.51 | 117.54 | 1.85 | 118.29 | 85.54 | 37.38 | 2.69 | 0.22 |
| #4 | 1845.05 | 0.10 | 770.72 | 181.54 | 4.25 | 138.48 | 84.76 | 26.05 | 2.76 | 0.14 |
| #5 | 1271.02 | 0.06 | 526.92 | 170.20 | 3.10 | 105.29 | 82.42 | 26.49 | 1.85 | 0.14 |
| #6 | 1663.77 | 0.10 | 593.45 | 308.56 | 1.92 | 150.85 | 84.99 | 32.31 | 2.81 | 0.18 |
| #7 | 1393.36 | 0.08 | 462.17 | 327.00 | 1.41 | 189.27 | 85.67 | 35.98 | 2.83 | 0.21 |
| #8 | 2087.42 | 0.10 | 646.57 | 318.21 | 2.03 | 209.70 | 86.05 | 32.25 | 3.18 | 0.18 |
| #9 | 936.13 | 0.03 | 398.71 | 92.86 | 4.29 | 196.41 | 85.23 | 36.50 | 2.69 | 0.21 |
| #10 | 1263.28 | 0.05 | 541.16 | 126.60 | 4.27 | 219.82 | 86.45 | 34.65 | 3.68 | 0.20 |
| Mean | 1366.92 | 0.07 | 513.57 | 197.39 | 2.89 | 161.12 | 85.19 | 32.73 | 2.85 | 0.18 |

### 3.2. Elevation Analysis

The DEM was reclassified into six categories at a classification interval of 40 m (Figure 6a) to analyze the elevation variation of different ridges inside and outside. Moreover, the frequency statistics of the elevation (Figure 6b) and the proportion of elevation areas inside and outside were analyzed (Figure 6c).

The minimum elevation value was 1543.29 m, the maximum value was 1769.66 m, and the mean value was 1647.01 m. The elevation was increased sharply to 1600 m after 1580 m. Subsequently, there was a significant valley close to 1660 m, which was the upper part of the outside in the northeast direction of the dividing line. The elevation primarily ranged from 1540 m to 1580 m, accounting for the highest proportion of 30.87%, which was mainly distributed outside. In general, the areas with the highest elevation were distributed on

both sides of the inside, and the elevation of the central area was distributed in the range of 1660–1700 m, accounting for 44.57%. The outside elevation tended to be decreased from north to south, and was mainly distributed in the range of 1580–1620 m, accounting for 45.17%. The elevation distribution of the ridge close to the south was relatively consistent.

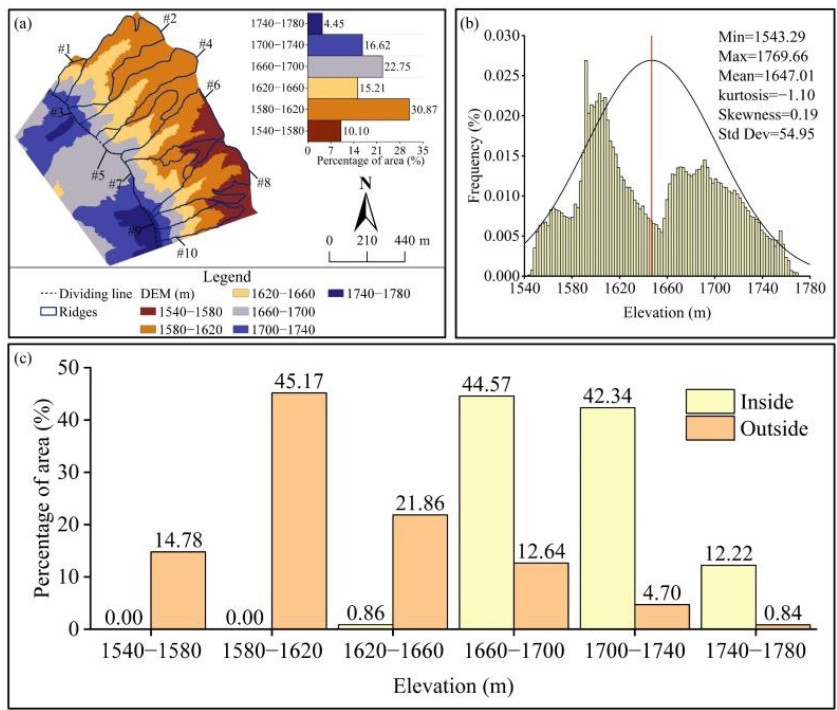

**Figure 6.** (**a**) The DEM reclassification results, (**b**) the elevation frequency statistics, and (**c**) the statistical results of the proportion of DEM areas inside and outside the pit rim structure.

### 3.3. Typical Feature Section Analysis

The section line analysis tool in ArcGIS was adopted to extract the longitudinal section and transverse section (Figure 7), so as to detect the change of topographic features at different locations. The longitudinal section line $AA'$ inside the pit rim structure served as the reference line, and $BB'$ and $CC'$ were constructed in parallel at 300 m intervals. Three transverse lines extended from the inside to the outside, and transverse lines $DD'$, $EE'$ and $FF'$ were set in parallel at 300 m intervals. Notably, the extension directions of the three transverse section lines and the three longitudinal section lines were perpendicular to each other.

#### 3.3.1. Longitudinal Analysis

The longitudinal section line $AA'$ indicated (Figure 8a) that the overall topography inside the pit rim structure was high on both sides and low in the center. A significant peak was identified on the left side, and the elevation change on the right side was faster. Moreover, the analysis was combined with the dividing line $MM'$ elevation inside and outside. To be specific, the red part indicated that $AA'$ was greater than the dividing line elevation and the blue part indicated that it was less than the dividing line elevation. The elevation change trend of $AA'$ was not significantly different from $MM'$. The left side was a continuous ridge extension, 8.68 m higher than the dividing line. The width of the ridge tended to be decreased, and the topography on both sides became steeper. The middle part was lower, thus forming a depression. At a distance of 630 m, the inside topography was uplifted, with an elevation of 39.42 m higher than the dividing line, and the topography fluctuated.

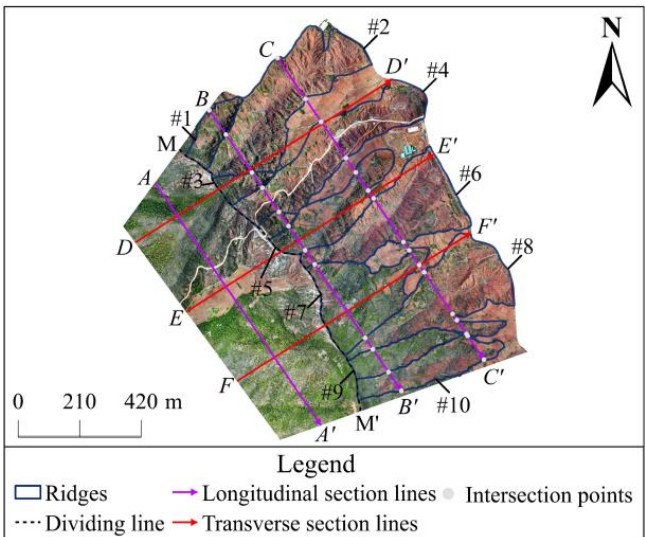

**Figure 7.** The distribution of the longitudinal and transverse section lines.

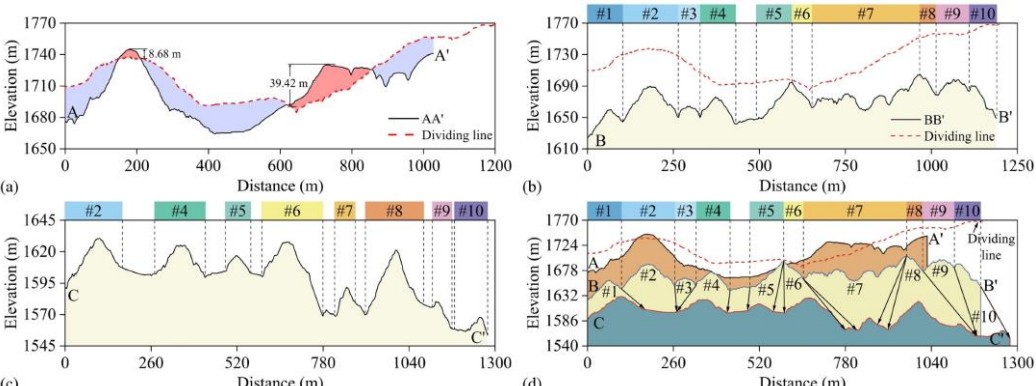

**Figure 8.** Results of the longitudinal section lines' elevation analysis, (**a**) the elevation change of the longitudinal section line $AA'$, (**b**) the elevation change of the longitudinal section line $BB'$, (**c**) the elevation change of the longitudinal section line $CC'$, and (**d**) the elevation change trend of the three longitudinal section lines.

$BB'$ was distributed on the upper part of the ridge outside, indicating the topography change of the upper (Figure 7). Most of the elevations of the upper part of the 10 ridges were lower than those of the dividing line (Figure 8b), whereas the elevation difference in the adjacent areas of ridges #5 and #6 was only about 3 m, and the topography change in this part and the dividing line were not significant. In general, the topography of the upper part fluctuated significantly, whereas the overall elevation change was not significant, and the elevation was roughly the same. With the increase of the horizontal distance, the maximum elevation of the ridges was decreased by nearly 130 m (Figure 8c), part of the ridges tended to disappear, and a gentle topography was formed between the ridges. The peak areas of the ridges became more significant, and the ridges on the left (#2, #4, #5, and #6) changed less significantly than the ridges on the right (#7, #8, #9, and #10). The width of ridge #7 varied the most, and the sides of ridge #8 tended to be steeper.

No significant difference was identified in the upper part of the ridge and the overall elevation trend (Figure 8d). As the ridges continued to extend, some of the smaller ridges disappeared, most of the ridges extended radially, the width of the ends of the ridges was increased, and gentle topography appeared between the ridges. It is noteworthy that ridges #7 to #10 changed most significantly. There was a mean elevation difference of nearly 48 m between $BB'$ and $CC'$, whereas that of ridges #1 to #6 was nearly 11 m.

### 3.3.2. Transversal Analysis

$d_1$, $d_2$, and $d_3$ represented the elevation variation of the mountain, as indicated by the transverse section line $DD'$. (Figure 9a). The minimum elevation at $d_2$ was 1724.27 m, and the peak elevation at $d_3$ was 1741.41 m. The horizontal distance from 453.78 m to 619.77 m ($d_4$ and $d_5$) was the topography change on the right side of the #2 ridge, and the topography behind the ridge end largely included flat farmland.

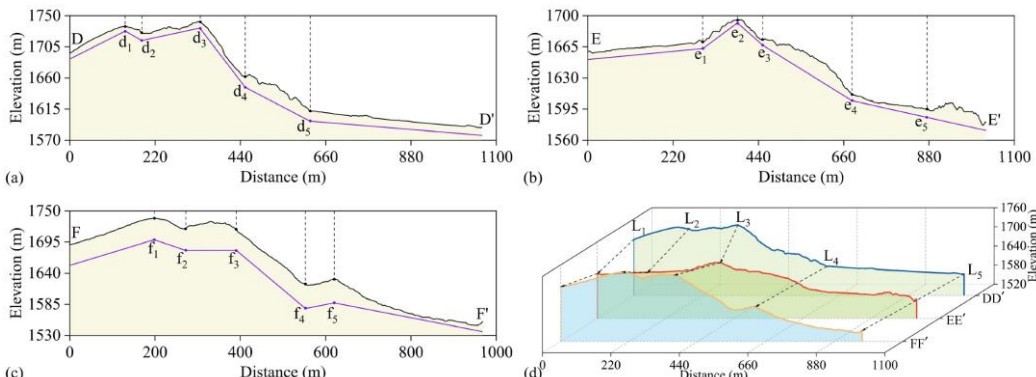

**Figure 9.** Results of the transverse section lines' elevation analysis, (**a**) the elevation change of the transverse section line $DD'$, (**b**) the elevation change of the transverse section line $EE'$, (**c**) the elevation change of the transverse section line $FF'$, and (**d**) the elevation change trend of the three transverse section lines.

$EE'$ mainly reflected the elevation change in the middle from the inside to the outside (Figure 9b). When the horizontal distance was less than 296.88 m ($e_1$), the topography changed gently. The horizontal distance was from 296.78 m to 452.77 m ($e_1$, $e_2$, and $e_3$), was higher ($e_1$ and $e_2$), and the elevation peak was 1695.16 m ($e_2$). The topography then decreased ($e_2$ and $e_3$). The topography changed of the #5 ridge originated from the horizontal distance of 450.97 m to 681.16 m ($e_3$ and $e_4$), and the topography changed of the end edge of the #6 ridge at the horizontal distance of 874.65 m ($e_5$). A gentle topography between ridges was between $e_4$ and $e_5$. $FF'$ indicated (Figure 9c) that the topography elevation inside the pit rim structure increased with a peak of 1737.62 m and a gentle change. The horizontal distance was 553.18 m to 620.37 m ($f_4$ and $f_5$), it was a plateau topography formed within the #7 ridge, followed by a sharp change.

The three transverse lines were combined and then analyzed in the form of a waterfall chart, the peak elevation of the three lines occurred inside, and the maximum elevation differences of the lines reached 153.41 m, 118.89 m, and 190.36 m. The variation trend of topography in the center was significantly lower than the two sides ($L_1$, $L_2$, and $L_3$), and the overall topography was depressed. The horizontal distance from 334.18 m to 620.97 m ($L_3$ and $L_4$) reflected the change in ridge topography, and no significant difference was identified in elevation. The horizontal distance from 620.97 m to 1062.25 m ($L_4$ and $L_5$) had no significant topography fluctuations. $DD'$ and $EE'$ intersected at a horizontal distance of 718.47 m. The elevation of the intersection point was 1605.79 m, and the subsequent topography change trend was the same. $DD'$ and $FF'$ intersected at a horizontal distance of 668.37 m, and the elevation of the intersection point was 1609.36 m. Although the topography had no significant fluctuation after the intersection point, the elevation of $FF'$ dropped sharply.

### 3.4. Extraction and Analysis of Valley Lines and Ridge Lines

The DEM was processed using a hydrological analysis method to automatically extract the valley and ridge lines, and the 5 m contour lines were superimposed to refine the extraction (Figure 10). The 3D display was achieved by overlaying the extraction results with the point-cloud for a multi-angle analysis of the orientation, and the results were projected to the XY, and XZ planes, respectively. In particular, the coordinates of the starting

and ending points of the valley lines and ridge lines were extracted in combination with the 3D model to determine the height difference and dip morphological parameters, and the trigonometric function calculation equation was employed for statistics.

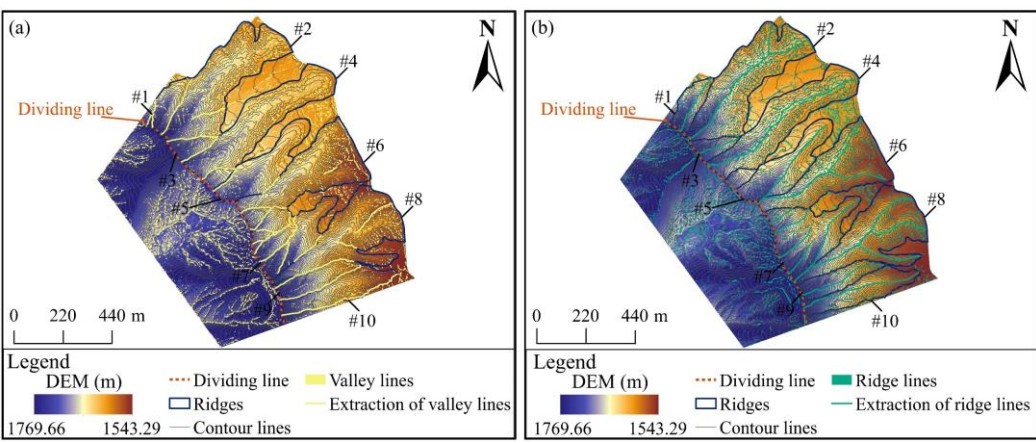

**Figure 10.** (**a**) The valley lines extraction results, and (**b**) the ridge lines extraction results.

### 3.4.1. Extraction and Analysis of Valley Lines

The valley lines inside the pit rim structure were primarily distributed on the mountains in the northwest and southwest (Figure 10a). A significant water flow path was formed due to the low topography in the central area. The extracted 23 main valley lines (G1–G23) were evenly distributed outside (Figure 11a), and most of the valley lines were consistent with the extracted ridge boundaries. The valley lines formed between the ridges on the left were shorter, and these lines were largely distributed in the upper part of the ridges. The valley lines on the right were more densely distributed and then extended to the bottom (Figure 11b), and the catchment area on the right was more significant. In general, the valley lines formed a significant vertical drop in a relatively short horizontal distance (Figure 11c).

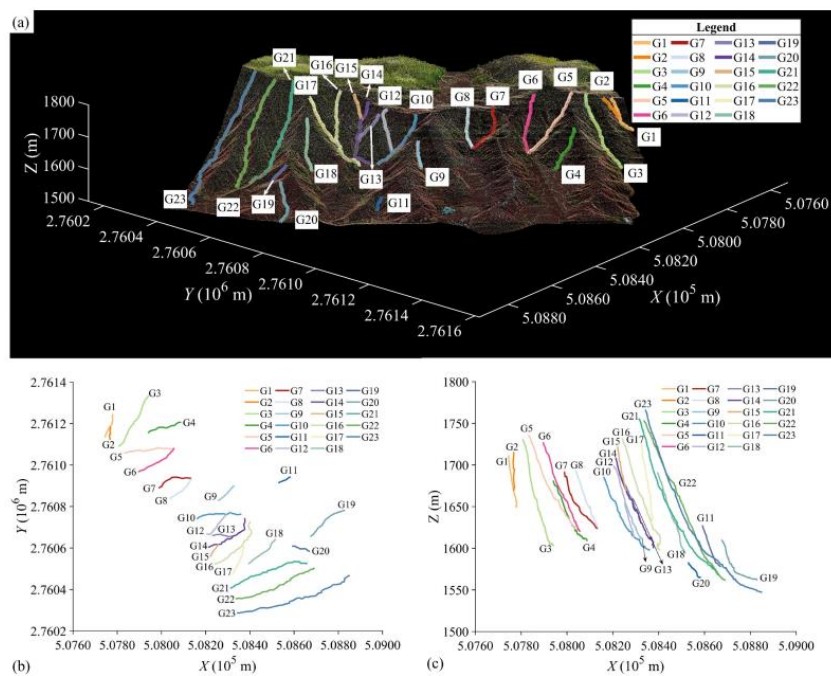

**Figure 11.** (**a**) The valley lines combined with point-cloud display, (**b**) the valley lines XY projection, and (**c**) the valley lines XZ projection.

According to statistics (Table 4), the mean length of the morphological parameters of the valley lines was determined as 202.53 m, the mean maximum elevation difference was 93.31 m, and the mean dip angle reached 29.37°. In general, the longer valley lines were distributed between the ridges at the outer two ends, and there were shorter valley lines in the central area. To be specific, G23 achieved the longest length of 581.88 m, a vertical drop of 219.17 m, as well as a relatively gentle dip angle. The valley lines (G20 and G11) close to the end of the ridge line achieved the smallest dip angle.

**Table 4.** Measurement results of valley lines' morphological parameters.

| No. | Length (m) | Maximum Elevation Difference (m) | Dip (°) |
|-----|-----------|----------------------------------|---------|
| G1  | 123.68 | 62.23  | 30.21 |
| G2  | 68.09  | 46.20  | 42.73 |
| G3  | 289.11 | 127.68 | 26.21 |
| G4  | 166.50 | 72.32  | 25.74 |
| G5  | 233.60 | 115.88 | 29.74 |
| G6  | 206.02 | 107.02 | 31.30 |
| G7  | 168.87 | 67.74  | 23.65 |
| G8  | 141.88 | 68.54  | 28.89 |
| G9  | 107.00 | 42.84  | 23.60 |
| G10 | 213.52 | 87.91  | 24.31 |
| G11 | 62.75  | 18.60  | 17.24 |
| G12 | 140.29 | 86.09  | 37.85 |
| G13 | 97.28  | 59.90  | 38.01 |
| G14 | 254.97 | 107.97 | 25.05 |
| G15 | 72.00  | 52.49  | 46.81 |
| G16 | 302.87 | 134.61 | 26.39 |
| G17 | 154.60 | 95.23  | 38.02 |
| G18 | 172.33 | 96.72  | 34.14 |
| G19 | 89.01  | 49.58  | 33.85 |
| G20 | 208.90 | 48.09  | 13.31 |
| G21 | 398.41 | 187.54 | 28.08 |
| G22 | 404.75 | 191.77 | 28.28 |
| G23 | 581.88 | 219.17 | 22.13 |
| Mean | 202.53 | 93.31 | 29.37 |

3.4.2. Extraction and Analysis of Ridge Lines

The valley lines inside the pit rim structure were primarily distributed on the mountains in the northwest direction (Figure 10b), and the 18 main ridge lines (J1–J18) were evenly distributed outside (Figure 12a). The left side was mostly a single ridge line extending from the highest to the bottom (Figure 12b), whereas the ridge lines on the right were densely distributed and diverged from the end to form shorter ridge lines connected with the main ridge lines. In general, the elevation change in the upper and middle parts of the ridge lines was significant, and the elevation change trend in the end area tended to be flattened (Figure 12c).

According to the statistics (Table 5), the mean length of the morphological parameters of the ridge lines reached 376.05 m, the mean maximum elevation difference was 116.77 m, and the mean dip angle was determined as 21.93°. The length of the ridge lines exceeded 110 m, the trend was relatively gentle compared with the valley lines, and the vertical drop distance was higher. The longest ridge line length of J5 reached 820.24 m, the maximum elevation difference was determined as 131.93 m, the dip angle was obtained as 9.26°, and the topography of the #4 ridge changed gently. The J3 ridge line was located on the #3 ridge, the maximum elevation difference was only 21.63 m, and the minimum dip angle was 6.51°. The J9 and J12 ridge lines were located in the #7 ridge, with a dip angle greater than 40°, and the #7 ridge was the steepest.

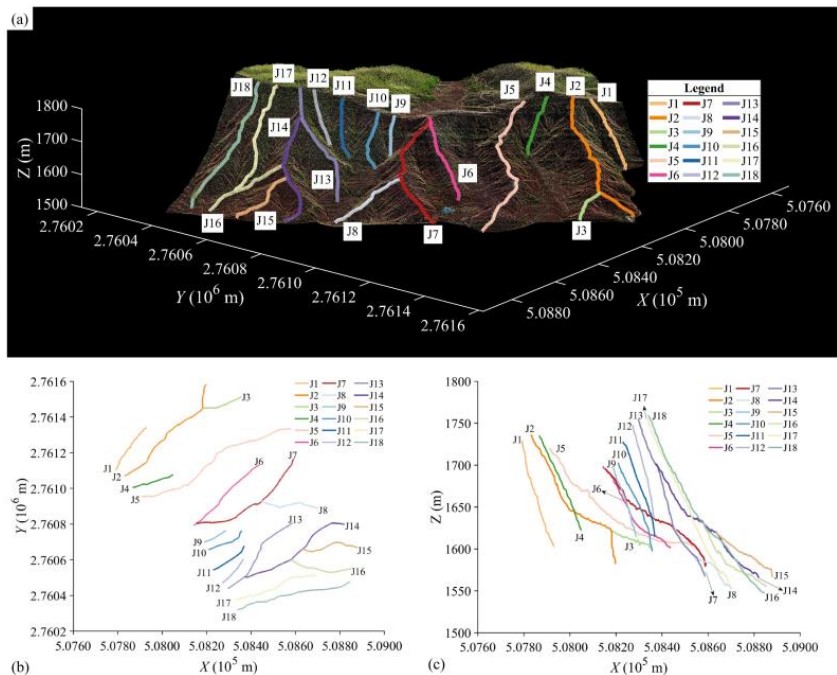

**Figure 12.** (**a**) The ridge lines combined with point-cloud display, (**b**) the ridge lines' XY projection, and (**c**) the ridge lines' XZ projection.

**Table 5.** Measurement results of ridge lines' morphological parameters.

| No. | Length (m) | Maximum Elevation Difference (m) | Dip (°) |
|-----|-----------|----------------------------------|---------|
| J1 | 283.89 | 126.33 | 26.42 |
| J2 | 670.46 | 153.70 | 13.25 |
| J3 | 190.67 | 21.63 | 6.51 |
| J4 | 199.09 | 112.78 | 34.50 |
| J5 | 820.24 | 131.93 | 9.26 |
| J6 | 453.79 | 96.99 | 12.34 |
| J7 | 617.12 | 115.50 | 10.79 |
| J8 | 279.38 | 73.42 | 15.24 |
| J9 | 118.04 | 80.74 | 43.16 |
| J10 | 195.22 | 104.54 | 32.38 |
| J11 | 198.31 | 112.97 | 34.73 |
| J12 | 166.40 | 111.05 | 41.86 |
| J13 | 485.33 | 187.01 | 22.66 |
| J14 | 571.26 | 136.09 | 13.78 |
| J15 | 271.41 | 53.99 | 11.47 |
| J16 | 294.22 | 76.19 | 15.01 |
| J17 | 410.09 | 195.79 | 28.52 |
| J18 | 543.99 | 211.29 | 22.86 |
| Mean | 376.05 | 116.77 | 21.93 |

### 3.5. Slope and Aspect Analysis

Slope and aspect analysis was conducted using DEM (Figure 13), and the slopes were reclassified into 10 classes with a 9° interval. The proportions of the slope and aspect areas of the whole pit rim structure were calculated using the method of mathematical statistics (Figure 13a,b), and the frequency distribution of slope and aspect was determined (Figure 13c,d). The statistical parameters comprised minimum, maximum, mean, kurtosis, skewness, and standard deviation.

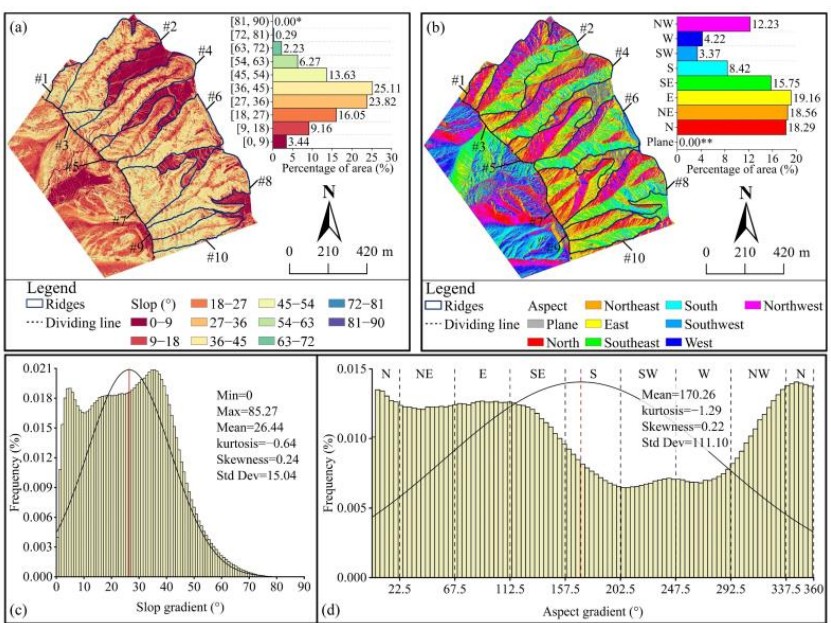

**Figure 13.** (**a**) Slope reclassification results, (**b**) aspect analysis results, (**c**) slope frequency statistics, and (**d**) aspect frequency statistics (* 0.0007181, ** 0.0001323).

### 3.5.1. Slope Analysis

The maximum value of the pit rim structure slope was 85.27°, the mean value was 26.44°, the kurtosis was −0.64, and the skewness was determined as 0.24. The data distribution was flatter than the normal distribution, and the distribution was positively skewed, such that a long tail was formed on the right side. The above result indicated that most slopes had extreme values on the right side, and the degree of dispersion was strong (Figure 13a,c). Most of the slopes primarily ranged from 27° to 45°, accounting for 48.93%, and they were distributed in the steep slope areas on both sides of the ridge. The first peak was identified between 0° and 9° in the slope, which was distributed in the flat farmland area formed outside, the second peak was identified between 42° and 54°, which was distributed on the upper part of the ridge. Notably, the classification results of the slope inside and outside were different. Accordingly, the dividing line was used for division, and the classification results of the slope inside and outside were determined (Table 6 and Figure 14a). Furthermore, the statistics of the slope change of the outside independent ridges were analyzed (Table 7 and Figure 14b).

**Table 6.** Statistical results of the proportion of the inside and outside slope area of the pit rim structure.

| Location | Percentage of Slop Area (%) | | | | | | | | | |
|---|---|---|---|---|---|---|---|---|---|---|
| | [0, 9) | [9, 18) | [18, 27) | [27, 36) | [36, 45) | [45, 54) | [54, 63) | [63, 72) | [72, 81) | [81, 90) |
| Inside | 14.78 | 24.81 | 24.36 | 17.90 | 11.26 | 4.92 | 1.60 | 0.34 | 0.03 | 0.00 * |
| Outside | 6.68 | 5.79 | 64.93 | 8.68 | 8.43 | 3.71 | 1.33 | 0.40 | 0.05 | 0.00 ** |

* 0.0002318, ** 0.0004261.

The slope inside was largely concentrated between 0° and 36°, of which the 9° to 18° gentle slope area accounted for 24.81%, mainly distributed in the pit rim structure edge (Table 6). The bottom of the depression was formed internally and the hilly area on both sides. For the area with a slope over 36°, the proportion of the area tended to be decreased, and the slope did not change significantly. The slope outside the pit rim structure was primarily concentrated between the steep slopes of 18° to 27°, accounting for 64.93% of the area, mainly distributed on both sides of the ridge. The area of the steep slopes with a

slope over 27° was the second, largely distributed in the upper part of the ridges close to the dividing line.

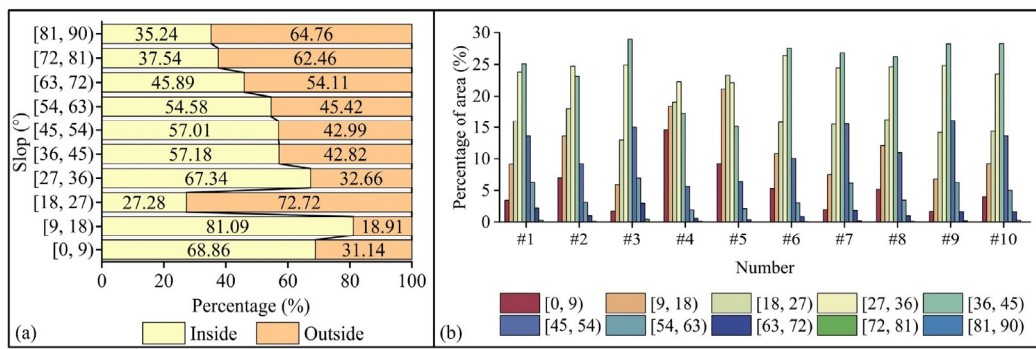

**Figure 14.** (**a**) Results of the percentage accumulation of the inside and outside of the slope of the pit rim structure, and (**b**) results of the percentage area of independent ridges' slope.

**Table 7.** Statistical results of the slope areas of independent ridges.

| Ridge ID | Slope Area (m²) | | | | | | | | | |
|---|---|---|---|---|---|---|---|---|---|---|
| | [0, 9) | [9, 18) | [18, 27) | [27, 36) | [36, 45) | [45, 54) | [54, 63) | [63, 72) | [72, 81) | [81, 90) |
| #1 | 766.20 | 2040.60 | 3576.64 | 5306.76 | 5594.60 | 3037.92 | 1397.20 | 496.84 | 63.56 | 0.16 |
| #2 | 8318.96 | 16,219.48 | 21,520.04 | 29,405.20 | 27,553.64 | 10,941.68 | 3694.44 | 1188.00 | 128.44 | 1.64 |
| #3 | 241.44 | 823.56 | 1808.32 | 3464.92 | 4022.12 | 2085.04 | 967.00 | 417.32 | 64.48 | 0.48 |
| #4 | 14,801.76 | 18,724.80 | 19,369.80 | 22,644.76 | 17,623.16 | 5722.96 | 1955.24 | 589.04 | 110.60 | 2.44 |
| #5 | 5301.20 | 12,128.56 | 13,384.12 | 12,716.88 | 8688.72 | 3668.20 | 1232.80 | 227.28 | 9.64 | 0.00 |
| #6 | 5070.80 | 10,404.60 | 15,183.20 | 25,327.96 | 26,420.76 | 9643.64 | 2920.72 | 846.08 | 85.20 | 0.16 |
| #7 | 1524.80 | 5811.36 | 12,001.48 | 18,934.56 | 20,777.56 | 12,052.44 | 4780.00 | 1425.08 | 145.12 | 0.20 |
| #8 | 5343.16 | 12,494.56 | 16,816.32 | 25,408.72 | 27,048.12 | 11,328.72 | 3591.36 | 1006.40 | 108.24 | 0.32 |
| #9 | 496.60 | 2018.12 | 4197.24 | 7348.16 | 8356.28 | 4778.08 | 1843.92 | 491.08 | 49.84 | 0.12 |
| #10 | 1819.28 | 4175.60 | 6489.92 | 10,638.80 | 12,789.32 | 6172.80 | 2263.68 | 748.44 | 121.04 | 2.40 |

The results of the percentage accumulation of the inside and outside slope are presented in Figure 14a. The proportion of 0° to 18° inside was larger than outside, and the inside largely comprised gentle depressions and hills. Since the outside was dominated by ridge–valley staggered topography, with a slope between 18° and 27°, the proportion of the outside exceeded that of the inside. The topography at both ends of the dividing line was uplifted upward, and the proportion of the inside between 27° and 63° exceeded that of the outside. However, when the slope reached over 63°, the proportion of the slope outside was higher than that of the inside, which was prone to extreme gradients.

To quantify the change in the slopes of the ten ridges, the proportions of the slope areas were calculated (Table 7 and Figure 14b). The slope distribution of the ridges (#6–#10) on the left side near the edge was relatively consistent, and no significant difference was identified in the overall topography, while the ridges on the left had various shapes. To be specific, the slopes of #7 of the #10 ridges were mainly concentrated in the range of 36–45° sharp slopes. The slopes of the #2 and #4 ridges were mainly concentrated in the range of 27–36° steep slopes. The two ridges extend from the highest to the bottom. No significant difference was identified in the ridge shape, and the end of the #4 ridge formed a gentle slope (0–9°) and had an area ratio which was the highest among all ridges. The #5 ridge was distributed in the upper, the end of the ridges did not extend to the bottom, and the slope was concentrated on the slope of 18–27°.

### 3.5.2. Aspect Analysis

Combined with the classification of the aspect (Figure 13b) and the statistical results of the aspect frequency (Figure 13d), the mean slope aspect was 170.26° (south direction), the

kurtosis was −1.29, and the skewness was determined as 0.22. The data distribution was significantly different from the normal distribution, the distribution pattern was positively skewed, and the slope aspect values in the right part were highly discrete. The pit rim structure was dominated by the aspects of the north, northeast, east, and northwest, and the east area accounted for the highest proportion of 19.16%. The aspect changed significantly in the south, west, and southwest directions, and the west and southwest aspects accounted for the least. In particular, the aspect distribution inside and outside was significantly different. Thus, the aspect distribution was determined using the same method as the slope analysis (Table 8 and Figure 15a). The aspect distribution of the outside ridges was quantitatively analyzed (Table 9 and Figure 15b).

**Table 8.** Statistical results of the proportions of the inside and outside aspect areas of the pit rim structure.

| Location | Percentage of Aspect Area (%) | | | | | | | | |
|---|---|---|---|---|---|---|---|---|---|
| | Plane | N | NE | E | SE | S | SW | W | NW |
| Inside | 0.00 * | 12.89 | 4.59 | 5.40 | 9.93 | 12.39 | 18.77 | 18.35 | 17.67 |
| Outside | 0.00 ** | 18.01 | 19.63 | 19.82 | 15.79 | 7.96 | 3.33 | 4.11 | 11.35 |

* 0.001444, ** 0.000898.

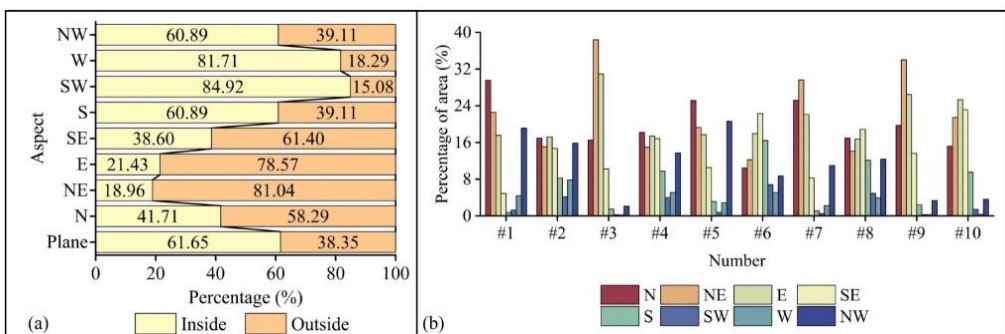

**Figure 15.** (**a**) Results of the percentage accumulation of inside and outside aspects of the pit rim structure, and (**b**) results of the percentage area of independent ridges' aspect.

**Table 9.** Statistical results of the aspect areas of independent ridges.

| Ridge ID | Aspect Area (m²) | | | | | | | | |
|---|---|---|---|---|---|---|---|---|---|
| | Plane | N | NE | E | SE | S | SW | W | NW |
| #1 | 0.01 | 6586.20 | 5028.36 | 3914.48 | 1088.00 | 154.36 | 271.72 | 979.80 | 4257.40 |
| #2 | 0.00 | 20,161.44 | 17,915.04 | 20,475.36 | 17,484.52 | 9828.68 | 4932.96 | 9321.76 | 18,851.72 |
| #3 | 0.00 | 2290.16 | 5329.04 | 4300.60 | 1421.40 | 207.28 | 32.76 | 24.04 | 291.00 |
| #4 | 0.01 | 18,483.92 | 15,158.84 | 17,698.52 | 17,095.16 | 9922.72 | 4056.36 | 5190.56 | 13,938.32 |
| #5 | 0.01 | 14,422.28 | 11,037.36 | 10,155.16 | 6055.44 | 1794.56 | 438.96 | 1636.84 | 11,816.48 |
| #6 | 0.00 | 10,045.24 | 11,741.88 | 17,222.00 | 21,378.36 | 15,751.04 | 6537.28 | 4878.08 | 8349.20 |
| #7 | 0.00 | 19,496.32 | 22,968.28 | 17,132.08 | 6399.96 | 840.28 | 362.96 | 1725.24 | 8527.40 |
| #8 | 0.00 | 17,487.48 | 14,534.44 | 17,289.08 | 19,436.40 | 12,519.68 | 5063.96 | 4088.32 | 12,726.52 |
| #9 | 0.00 | 5817.92 | 10,059.12 | 7825.68 | 4031.08 | 703.68 | 75.68 | 80.64 | 985.64 |
| #10 | 0.00 | 6870.80 | 9685.80 | 11,449.72 | 10,466.28 | 4309.24 | 642.64 | 159.76 | 1637.00 |

The proportion of the southwest aspect inside was highest at 18.77%, the outside northeast aspect proportion was highest at 19.63%, and the two aspect distributions were opposite (Table 8). The inside was mainly distributed in the south, southwest, west, and northwest directions (Figure 15a), and the vegetation coverage was high. However, the distribution of the aspect outside was the opposite, and there was a significant difference in the distribution of solar thermal resources. Since the inside was mainly depression topography, the aspect of the plane was higher than the outside.

The aspect of the ridges was largely concentrated in the clockwise direction from north to the southeast (Figure 15b), and the maximum value of the ridges was evenly distributed (Table 9). The plane aspect was the lowest, mainly distributed on the #1, #4, and #5 ridges. Most of the ridges were northwest, whereas ridges #3, #9, and #10 were not dramatically increased. The distribution of the aspect of the #2, #4, and #8 ridges and the distribution of the #1 and #5 ridges were relatively similar. Although the maximum value of the aspect of the different ridges outside was different, a certain law was maintained in the distribution of the aspect. The aspect was primarily distributed in the north, northeast and east, and the east and northeast accounted for the highest proportion. The lowest proportion was in the west and southwest, whereas the proportion in the northwest tended to be increased again. The ridges in the northern part of the pit rim structure (#1–#5) primarily accounted for a greater proportion of northwest aspect than the ridges in the south.

### 3.6. TWI Feature Analysis

3.6.1. Statistical Analysis of TWI Classification

The TWI can be used to identify water flow paths, with low values representing bulges and high values representing depressions, forming troughs [62,63]. The change in the static soil water content may lead to a change in the soil strength, thus affecting the stability of the ridge. Accordingly, to explore the hydrological path relationship at different positions of the pit rim structure, TWI analysis was conducted (Figure 16a,b).

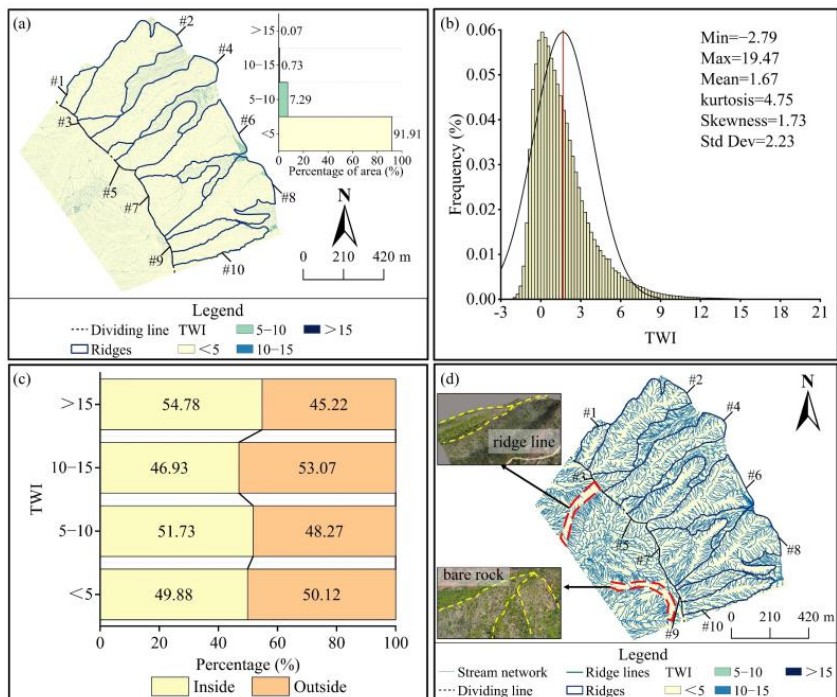

**Figure 16.** (**a**) TWI classification results, (**b**) TWI frequency statistics, (**c**) TWI area percentage accumulation statistics results inside and outside, and (**d**) TWI combined with river network and ridge lines analysis results.

The minimum value of TWI was determined as −2.79, the maximum value was 19.47, the mean value was 1.67, and the kurtosis was obtained as 4.75 (Figure 16b). The statistical results on the right had strong dispersion. The TWI was reclassified into four categories (Figure 16a), and the proportion of the TWI was less than 5, accounting for 91.92%, followed by 5–10. Moreover, the difference in TWI distribution was small, as indicated by the statistical results of the TWI area (Table 10) and the percentage accumulation (Figure 16c) inside and outside. The high value of TWI was largely obtained in the area where low-lying depressions and the outer ridges and valleys intersected. The catchment area in this area

was increased, especially the valleys formed between the ridges outside the structure. Moreover, the surface runoff on both sides of the ridges was rich, and the soil was easy to be saturated and runoff. In contrast, the areas beyond the catchment line achieved anomalously low values.

**Table 10.** Statistical results of the proportions of the inside and outside TWI areas of the pit rim structure.

| Location | Percentage of TWI Area (%) | | | |
|---|---|---|---|---|
| | <5 | 5–10 | 10–15 | >15 |
| Inside | 347,990.75 | 29,010.75 | 2539.00 | 293.5 |
| Outside | 754,893.75 | 58,443.75 | 6198.00 | 523.00 |

3.6.2. Exploration of TWI-based River Network Correlation

Although the TWI results indicate that the surface runoff was abundant, the distribution of river networks was not fully revealed. Thus, based on the TWI results combined with hydrological analysis methods, the river network was extracted, and the superimposed analysis was conducted in combination with the ridge lines (Figure 16d). Most of the extracted river network was consistent with the high-value areas in TWI, the river network inside was primarily distributed in the high-lying area, and the low-lying area was formed in the center. The river network in the central low-lying area was denser than the distribution on both sides, and differences were identified in the direction of the river network. No significant river network was formed on the ridge of the left mountain and the surface of the exposed rock layer on the right. Affected by soil erosion and rain erosion, the river network outside was mainly distributed on both sides of the ridge, forming a closed catchment line at the end of each ridge. The catchment area was formed by the circulation through the flat topography area that was formed between the ridges, and the direction of the river network formed by different ridges was not significantly different.

**4. Discussion**

*4.1. Geomorphological Application of Multi-Source Data*

The 3D model covers real color information, and is capable of visually indicating the structural features. Moreover, it has coordinated projection information, such that the fast and quantitative acquisition of different geomorphic structure feature information can be achieved from multiple angles of points, lines, and surfaces. Sheng, et al. [64] achieved qualitative and quantitative section interpretations, analysis, and measurements using 3D models of high-slope outcrop landforms. Huang, et al. [65] used a 3D model of the dangerous rock mass of high and steep slopes to realize an analysis of the spatial distribution, instability mode, and evolution process. In addition, compared with the dense point-cloud of UAVs, the accuracy of the point-cloud obtained by airborne LiDAR is higher [66].

Compared with the conventional use of DEM for analysis, through different DEM visualization methods, different geographical features can be differentiated, and detailed information can be highlighted. Wang, et al. [67] used the red relief image map (RRIM) processing method to interpret different geological hazards and landforms. Favalli and Fornaciai [68] compared the different visualization methods to achieve optimal volcano identification and division of specific areas. Chen, et al. [69] used a variety of visualization methods to perform a 3D analysis of DEM and effectively identified different landforms.

Based on the comparison of a wide variety of visualization methods, the SIM and SVF visualization methods can more effectively indicate the geomorphology and more clearly display features (e.g., steep slopes, bedding, and outcrop landforms). Moreover, the integrated exploration of the pit rim structure with multi-source data was achieved using a 3D model, airborne LiDAR point-cloud, and the results of visual analysis (Figure 17).

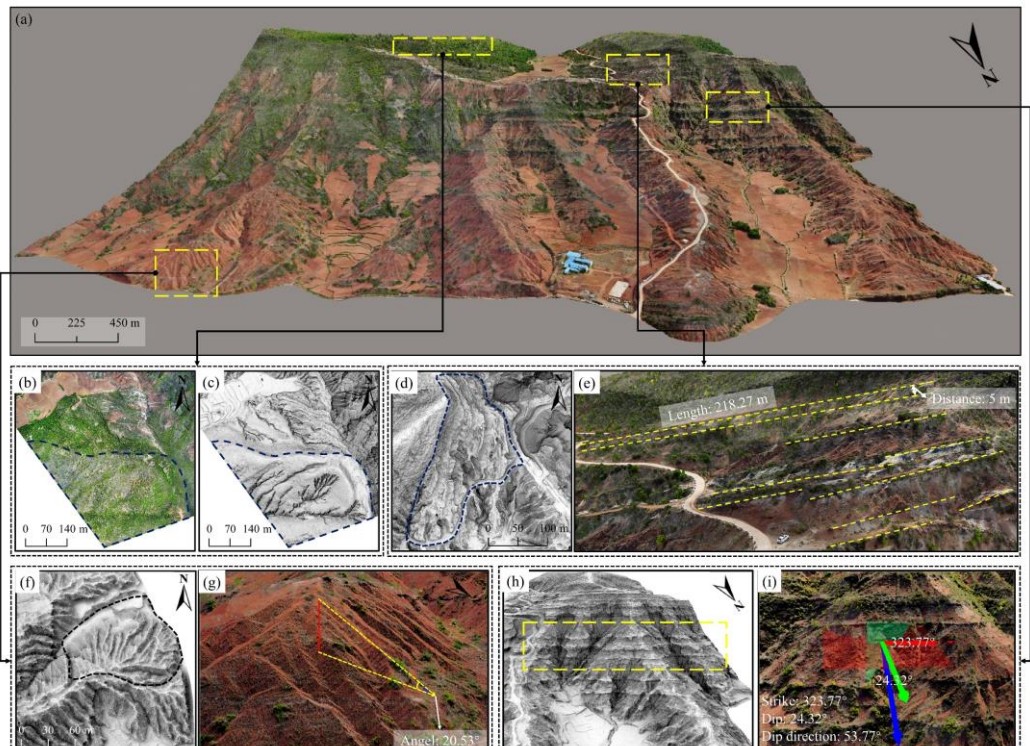

**Figure 17.** (**a**) A 3D model of the pit rim structure, (**b**) DOM of the gully inside the pit rim structure, (**c**) SVF visualization result of the gully, (**d**) SIM visualization result of the outcrop, (**e**) a 3D model of the outcrop's geomorphological measurement, (**f**) SVF visualization results of the ridge end, (**g**) ridge slope measurement based on the LiDAR point-cloud, (**h**) SIM 3D bedding visualization results, and (**i**) bedding strike, dip and dip direction measurements based on the LiDAR point-cloud (the green arrow represents perpendicular to the strike line).

Four representative geographical features were selected for analysis based on the 3D model. The DOM and SVF visualization results of the gully inside are presented in Figure 17b,c. The geomorphic of the mountain's right side was distributed in a stepped distribution (Figure 17d). This part primarily comprised the outcrops with a staggered distribution (Figure 17e). With the use of the 3D model, outcrops at different locations were effectively identified, and the length, spacing, and trend of the outcrops was obtained simultaneously [70]. The SVF visualization results (Figure 17f) highlighted the hydrological path formed by soil erosion at the end of the ridges. Combined with the LiDAR point-cloud, the dip angle of each river network was accurately obtained (Figure 17g). The 3D display of the SIM visualization results effectively highlighted the strip-like bedding structure formed in the middle of the mountain (Figure 17h). The strike, dip, and dip directions between beddings were accurately measured (Figure 17f).

### 4.2. Effect of Data Collection Methods and Results Accuracy

4.2.1. Effect of Data Collection Methods

The route planning takes on a particularly critical significance, affecting the integrity of the data and the accuracy of the results. The zigzag route planning method has been used as one of the main methods, and is more applicable to data collection by rotary-wing UAVs [71]. However, the image resolution and the overlap between images will be significantly different for areas with significant vertical gradient differences. For vertically distributed landforms, route planning methods including nap-of-the object photogrammetry [72], or layered layout [73] can be used, or even manual flight [22] for data acquisition. The above methods are subject to the problem of redundancy. In this study, the ground-based flight route planning method was applied. The measurement accuracy analysis of LiDAR point-

cloud and 3D model results indicated that this method ensured the integrity of data in areas with large topographic fluctuations, and DEM accurately indicated the topography.

However, when collecting field data, a single route type is not necessarily suitable for all landforms. Furthermore, the optimization of route planning parameters (e.g., flight height, lens angle [74], and overlap) should be fully considered in accordance with the actual topographic characteristics. Combined with multi-source data collection methods [75], ensuring the acquisition of high-quality and high spatial resolution results has significant advantages for multi-type and multi-scale identification and analysis.

4.2.2. Effect of Results Accuracy

The multi-perspective topography features were identified by combining the high-resolution DEM and the 3D model that restored the real topography. Visualization analysis, related topography feature analysis, and mathematical and statistical methods were combined for measurement and analysis. However, DEM with different resolutions may affect the analysis results. For environments with significant vertical differences, the possibility of gradual smoothing of topography feature parameter results increases with low DEM resolution [76].

Compared with the widely used DEM (e.g., 30 m, 25 m, and 1.5 m resolution), the DEM with 0.2 m resolution in this study was able to retain more subtle changes in topographical features and to achieve high precision based on the assurance of data processing efficiency. The topographical features parameter extraction was more reliable, and the data results were more reliable. However, the use of a single resolution was relatively limited. In future research, a differentiated comparative analysis can be conducted on the accuracy of topographical features parameter extraction using different high-resolution DEMs, and the optimal resolution DEMs can be selected to conform to the needs of different studies in accordance with the meanings and demands of different topography feature parameters.

*4.3. The Shortcomings of This Method*

In this study the real 3D model of a pit rim structure was rebuilt by combining UAV–LiDAR technology and UAV images, and a 3D point-cloud was obtained using LiDAR to rapidly build a high-precision DEM. The qualitative and quantitative geomorphological feature analysis was visualized and refined using a 3D model and high-precision DEM. However, the following three problems remain. (1) The UAV–LiDAR system requires considerable time to construct a 3D model using UAV images, and the quality of local edge areas is poor. The problem of further optimizing the route planning parameters or integrating a 3D point-cloud to reduce data redundancy and improve the efficiency and quality of large-scale 3D model reconstruction should be studied and validated in depth. (2) The distribution of vegetation also makes a difference to the distribution of geomorphic types and topographic features, and vegetation point-cloud is not easily adopted due to the lack of information on vegetation types and basic features in the study area. Moreover, the unpublished hydrogeological, geological structure and fault data are key analysis factors. If these parts of the data are combined, the richness and accuracy of geomorphological feature analysis will be further increased. (3) Theoretically, this method can be employed for geological hazard investigation and monitoring as well as geomorphological evolution analysis in the same complex environment, whereas the specific applicability should be practiced in depth.

**5. Conclusions**

Based on the UAV–LiDAR system, the method of ground-based flight route planning was adopted in this study to obtain high-resolution images and high-precision LiDAR point-cloud data of the pit rim structural in a mountainous environment. Based on LiDAR point-cloud data processing, a high-precision DEM with a resolution of 0.2 m was built, and a 3D model was built using SfM–MVS technology. The multi-angle, qualitative, and quantitative topography feature identification, measurement, and analysis were achieved

through visual interpretation, DEM visualization, GIS topographic feature extraction, analysis, and other methods. The aim of this study was to gain insights into and quantify the ability of the UAV–LiDAR method to identify and analyze topographic features in high-altitude and high-drop plateau mountain environments. The main conclusions were drawn as follows:

(1) The UAV–LiDAR system adopted the ground-based flight route planning method, which was applicable to the complex topography environment. The elevation root mean square error of the 0.2 m high-resolution DEM from LiDAR was determined as 4.4 cm. The horizontal and elevation accuracy of the 3D model reached 4.4 cm and 7.9 cm, respectively. The data acquisition technology process based on UAV–LiDAR can provide a reference for obtaining high-quality data results.

(2) The SIM and SVF visualization methods can highlight the topography features, extracting the dividing line between the inside and the outside. In general, the elevation ranged from 1540 m to 1580 m, mostly distributed outside, and the elevation tended to be decreased from north to south. The inside of the elevation reached 1660 m to 1700 m, and the central topography was the lowest. The mean maximum elevation difference of the ten ridges was determined as 161.12 m, and the mean maximum slope was 81.59°.

(3) The 0.2 m high-resolution DEM can retain the topographic details and provide basic data for refined section analysis. Most of the topography inside the pit rim structure was lower than the elevation of the dividing line, whereas the topography on both sides was the opposite. No significant difference was identified in the upper ridges. A gentle topography was obtained with the continuous extension of the ridges. The main change was at the end of the right ridge.

(4) The mathematical statistics method can effectively analyze the results of the slope, aspect, and TWI topography feature parameters. The slope primarily ranged from 36° to 45°. The aspects inside and outside were opposite, and the maximum slope value was more likely to occur outside. The aspect was largely east, and significant differences were identified in the distribution of light and heat resources. Moreover, there was abundant surface runoff, and no significant difference was identified in the distribution of TWI values inside and outside. The valleys formed between the outside ridges and on both sides of the ridges were rich in surface runoff, and soil erosion was significant.

(5) The UAV–LiDAR system can provide fine basic data for the research on micro-topographic feature identification, measurement, and analysis. It has unparalleled advantages in extracting topographic feature information in the plateau and mountainous environment. The refined measurement and analysis will contribute to the further analysis of the correlation mechanisms of the formation and evolution of the topographical features and provide an important reference for the realization of high-precision 3D reproduction and quantitative analysis of micro-topography.

**Author Contributions:** R.B. performed the research and methodology, analyzed the data, and wrote the manuscript. S.G. (Shu Gan) designed the framework of the research and mastered the conceptualization. S.G. (Shu Gan) and X.Y. gave many suggestions for improving and modifying this paper. R.L., M.Y. and W.L. participated in data collection and investigation. S.G. (Sha Gao) and L.H. contributed to data processing and visualization analysis. All authors have read and agreed to the published version of the manuscript.

**Funding:** This research received the National Natural Science Foundation of China (No. 41861054 and No. 62266026) and the Yunnan Fundamental Research Project (No. 202201AU070108).

**Institutional Review Board Statement:** Not applicable.

**Informed Consent Statement:** Not applicable.

**Data Availability Statement:** The data are not publicly available as they involve the subsequent application of other studies.

**Conflicts of Interest:** The authors declare no conflict of interest.

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
