# Peer review of "Multi-View Analysis of High-Resolution Geomorphic Features in Complex Mountains Based on UAV–LiDAR and SfM–MVS: A Case Study of the Northern Pit Rim Structure of the Mountains of Lufeng, China"

_applsci, doi:10.3390/app13020738_

Round 1

Reviewer 1 Report

By adopting UAV-LiDAR and SfM-MVS methods to obtain the topographic features of a pit rim structure in Lufeng, Yunnan, China, as well as combining with a series of algorithm and mathematical statistic methods to process obtained data, the present work demonstrates a multi-view analysis of geomorphic features in complex mountain areas. This study is rich in content, but is a little verbose in expression. The comments that should be addressed are listed as follows:

1. While the writing is OK, definitely a proof reading is needed. (eg, the sentence in line 87)

2. In the introduction, a lot of methods with high-quality to obtain the geomorphological information were introduced, while in the present study the UAV-LiDAR method was adopted. What is the advantage of UAV-LiDAR methods compared with these introduced methods?

3. Line 79, the author said the accuracy of UAV images will reduce under the condition of geomorphic environment with high vegetation coverage, and from the demonstrated figures of the study area, the inside of the pit rim structure is high in vegetation coverage. How to ensure the accuracy of the geomorphic features from the obtained images of this part?  

4. The subsection 2.2.3 should be changed into 2.2.2.

5. Figure 2, the captions of figure 2 (c) and (e) are reversed.  

6. Line 291, from the data listed in table 3, the mean area is 0.07 km2.

7. Line 372, this description is incorrect, the elevation from e1 to e3 is increased first and then decreased, which is different from rose on both sides.

8. Figure 10, change Ectraction to Extraction. Besides, it is difficult to identify the dividing line in this figure, it is suggested to change the color of the dividing line to make it more visible for readers. 

9. Line 540, what is the law of the distribution of the aspect?

10. Correct the number of the fourth conclusion from 6) to 4).

Author Response

Dear Reviewer,

We deeply appreciate the effort and time you’ve spent in reviewing our manuscript “Multi-view Analysis of High-resolution Geomorphic Features in Complex Mountains Based on UAV-LiDAR and SfM-MVS: A Case Study of The Northern Pit Rim Structure Mountains of The Lufeng, China” (ID: applsci-2122716). We studied the comments carefully and have made revisions which we hope meet with your approval. The outline of the revisions is listed as follows:

  • We provide a rationale for why we chose the UAV-LiDAR system. (Please check Section Introduction)
  • We have added clarifications where there are unclear explanations in the text. (Please check the whole manuscript)
  • We add the shortcomings of the UAV-LiDAR system in the paper, and also include the shortcomings of the related methods used in this paper. (Please see Section Discussion)
  • We have corrected figures that were not displayed in the text and notes that were in the wrong order.
  • We have checked the experimental results of the whole manuscript carefully.
  • We have modified the format of the whole manuscript seriously.
  • We revised the whole manuscript carefully and modified the grammar and syntax errors. Meanwhile, we asked some colleagues who are skilled in English to help us for checking the language. We hope that the language is now acceptable for the next review process.

The revised parts are written in red text and annotations in the revised manuscript, and the detailed responses to the comments raised by the reviewers are as follows.

Comment #1

While the writing is OK, definitely a proof reading is needed. (e.g., the sentence in line 87).

Response:

Thank you for this comment. We have corrected the expressions and grammar of the whole paper. We have also added clarifications where the presentation is unclear.

Comment #2

In the introduction, a lot of methods with high-quality to obtain the geomorphological information were introduced, while in the present study the UAV-LiDAR method was adopted. What is the advantage of UAV-LiDAR methods compared with these introduced methods?

Response:

Thanks for your helpful comment. Based on the analysis of the various methods, we have added a description of why the UAV-LiDAR system is chosen in this paper. On the one hand, we hope to construct the 3D model through UAV images to realize all-around contactless geomorphic feature identification and quantitative analysis. On the other hand, to construct the high-resolution DEM for topographic feature analysis by discharging the influence of vegetation through LiDAR point cloud, which can circumvent the problem of insufficient data quality and provide multiple sources and high accuracy data for analysis through the mutual complement of the two data. Please refer to the lines 101 to 119 on the page 3.

Comment #3

Line 79, the author said the accuracy of UAV images will reduce under the condition of geomorphic environment with high vegetation coverage, and from the demonstrated figures of the study area, the inside of the pit rim structure is high in vegetation coverage. How to ensure the accuracy of the geomorphic features from the obtained images of this part?

Response:

Thanks for your helpful comment. We have added clarifications where the presentation of the section is unclear. The main reason for this is that when traditional UAV route planning methods are used, it may lead to a reduction in the quality of UAV images, which further leads to a reduction in the quality and accuracy of the 3D model. Since the experimental equipment for data acquisition is an ortho camera view. Therefore, in the data acquisition part, we used the ground-like flight route method to eliminate the problem of poor image quality in areas with high vegetation cover as much as possible. The accuracy of the acquired geographical features is ensured. Please refer to the line 78 on the page 2.

Comment #4

The subsection 2.2.3 should be changed into 2.2.2.

Response:

Thank you for this comment. We have corrected the incorrect section numbers. Please refer to the line 170 on the page 5.

Comment #5

Figure 2, the captions of figure 2 (c) and (e) are reversed.

Response:

Thank you for this comment. We have corrected the disordered statements. Please refer to the line 180 on the page 5.

Comment #6

Line 291, from the data listed in table 3, the mean area is 0.07 km2.

Response:

Thank you for this comment. We corrected the incorrect data and also checked the experimental data results in the whole paper. Please refer to the line 313 on the page 9.

Comment #7

Line 372, this description is incorrect, the elevation from e1 to e3 is increased first and then decreased, which is different from “rose on both sides”.

Response:

Thank you for this comment. We have corrected and added clarification to the incorrect statement in this section. Please refer to the line 393 on the page 12.

Comment #8

Figure 10, change “Ectraction” to “Extraction”. Besides, it is difficult to identify the dividing line in this figure, it is suggested to change the color of the dividing line to make it more visible for readers.

Response:

Thank you for this comment. We have corrected the incorrect English in Figure 10 and also redrawn the figure where it is not clearly shown. Please refer to the line 428 on the page 13.

Comment #9

Line 540, what is the law of the distribution of the aspect?

Response:

Thank you for this comment. We have added an explanation to this section. Please refer to the line 567 on the page 19.

Comment #10

Correct the number of the fourth conclusion from “6)” to “4)”.

Response:

Thank you for this comment. W We have corrected the incorrect serial number. Please refer to the line 748 on the page 23.

Reviewer 2 Report

(1) The English academic writing could be further improved, such as full writing of abbreviations of GIS (Line 22 ). 

(2) The authors should clearly explain the propose of the case study, such as how the results helped to analysis the land use type distribution, disaster identification, and geological movement of the study area in Lufeng. And why you chose to use UAV-LiDAR? 

(3) The disadvantages of UAV-LiDAR system should also be presented.

(4) Does the proposed analysis method have other potential application fields? 

Author Response

Dear Reviewer,

We deeply appreciate the effort and time you’ve spent in reviewing our manuscript “Multi-view Analysis of High-resolution Geomorphic Features in Complex Mountains Based on UAV-LiDAR and SfM-MVS: A Case Study of The Northern Pit Rim Structure Mountains of The Lufeng, China” (ID: applsci-2122716). We studied the comments carefully and have made revisions which we hope meet with your approval. The outline of the revisions is listed as follows:

  • We provide a rationale for why we chose the UAV-LiDAR system. (Please check Section Introduction)
  • We have added clarifications where there are unclear explanations in the text. (Please check the whole manuscript)
  • We add the shortcomings of the UAV-LiDAR system in the paper, and also include the shortcomings of the related methods used in this paper. (Please see Section Discussion)
  • We have corrected figures that were not displayed in the text and notes that were in the wrong order.
  • We have checked the experimental results of the whole manuscript carefully.
  • We have modified the format of the whole manuscript seriously.
  • We revised the whole manuscript carefully and modified the grammar and syntax errors. Meanwhile, we asked some colleagues who are skilled in English to help us for checking the language. We hope that the language is now acceptable for the next review process.

The revised parts are written in red text and annotations in the revised manuscript, and the detailed responses to the comments raised by the reviewers are as follows.

Comment #1

The English academic writing could be further improved, such as full writing of abbreviations of GIS (Line 22).

Response:

Thank you for this comment. We have corrected the expressions and grammar of the whole paper. We have also added clarifications where the presentation is unclear.

Comment #2

The authors should clearly explain the propose of the case study, such as how the results helped to analysis the land use type distribution, disaster identification, and geological movement of the study area in Lufeng. And why you chose to use UAV-LiDAR?

Response:

Thanks for your helpful comment. Based on the analysis of the various methods, we have added a description of why the UAV-LiDAR system is chosen in this paper. On the one hand, we hope to construct the 3D model through UAV images to realize all-around contactless geomorphic feature identification and quantitative analysis. On the other hand, to construct the high-resolution DEM for topographic feature analysis by discharging the influence of vegetation through LiDAR point cloud, which can circumvent the problem of insufficient data quality and provide multiple sources and high accuracy data for analysis through the mutual complement of the two data. Please refer to the lines 101 to 119 on the page 3.

I am sorry, probably because the current preliminary study focuses on the geomorphological identification and topographic features of the study area to understand the characteristics of the Lufeng area from the perspective of topographic features. And the subsequent analysis will further carry out studies on land use type distribution, hazard identification, geological movement, etc. for some specific areas according to the differentiation in the distribution of topographic features.

Comment #3

The disadvantages of UAV-LiDAR system should also be presented.

Response:

Thanks for your helpful comment. For this part of the narrative, we summarize and analyze the new section on methodological shortcomings in the discussion. On the one hand, it is an analysis of the shortcomings of the UAV-LiDAR system in the process of data acquisition and data processing. On the other hand, it is a summary of the shortcomings and the need for further improvement in the topographic feature analysis method. Please refer to the lines 691 to 709 on the pages 22 to 23.

Comment #4

Does the proposed analysis method have other potential application fields?

Response:

Thank you for this comment. For this part of the narrative, we summarize and analyze the new section on methodological shortcomings in the discussion. Please refer to the lines 691 to 709 on the pages 22 to 23.

To a certain extent, the analysis of the problem is lacking. I am sorry that probably due to the limited degree of the papers read, after reading the related papers on geohazard monitoring, geomorphological feature change analysis, and wetland change analysis, it was found that most of the methods were applied by a single UAV technology or LiDAR technology, and most of them used the DEM obtained from both technologies for the topographic analysis. The use of 3D models has not been fully exploited. This is one of the problems of having rich data results but a low usage rate.

Theoretically, the method used in this paper has a certain potential for the above-mentioned research directions, but the key is how to integrate the data results of the two techniques and use them together. Previous studies on mudslides and landslides have been carried out partly by UAV technology, but they are limited by the current research direction. There is an opportunity to work on mudflow and landslide hazard monitoring to explore the applicability and potential of this method.
